# A New Method for Collecting Large Amounts of Symbiotic Gastrodermal Cells from Octocorals

**DOI:** 10.3390/ijms21113911

**Published:** 2020-05-30

**Authors:** Hsiang-Yi Chiu, Li-Yi Lin, Ying Chen, En-Ru Liu, Hsing-Hui Li

**Affiliations:** 1Taiwan Coral Research Center, National Museum of Marine Biology and Aquarium, Pingtung 94450, Taiwan; chemistry_1028@gmail.com (H.-Y.C.); lly7997s@gmail.com (L.-Y.L.); 2Department of Oceanography, National Sun Yat-Sen University, Kaohsiung City 80424, Taiwan; louise120252@gmail.com; 3Department of Marine Biotechnology, National Kaohsiung University of Science and Technology, Kaohsiung 81143, Taiwan; gloria0963588082@gmail.com; 4Graduate Institute of Marine Biotechnology, National Dong Hwa University, Pingtung 94450, Taiwan

**Keywords:** high salinity, *Sinularia flexibilis*, osmotic pressure, cell viability, cell dissociation

## Abstract

The study of cnidarian-dinoflagellate endosymbiosis in octocorals is becoming increasingly important. As symbiotic gastrodermal cells (SGCs) are the key cells in a symbiotic relationship, obtaining SGCs and studying their functions represent an urgent need. The majority of the cells dissociated from octocoral tissues consist of host cells and algal cells, and very few intact SGCs can be observed. To solve this problem, we developed a new method to collect large amounts of SGCs from octocorals. We incubated the tissue of *Sinularia flexibilis* in high-salinity (60‰) filtered seawater for 6 h and were able to collect more than 18 times the number of SGCs from the control group. To test the quality of the dissociated cells, we performed three assays to evaluate their cell viability. All three assays demonstrated that cell viability was good after incubating in a high-salinity solution. We also used two other octocorals, *Paralemnalia thyrsoides* and *Sinularia compressa*, to perform the same experiment, and the results were similar to those for *Sinularia flexibilis*. Therefore, a high-salinity-induced increase in the SGC ratio is a common phenomenon among octocorals. This method allows researchers to collect large amounts of SGCs from octocorals and helps us to better understand the complex molecular interactions in cnidarian-dinoflagellate endosymbiosis.

## 1. Introduction

Coral reefs are the most diverse and productive marine ecosystems on the planet, providing ecosystem services that are vital to human societies [1]. In the last fifty years, global warming and ocean acidification have exacerbated local stresses from declining water quality and overexploitation of key species, driving the increasing scarcity of corals in reefs systems [2]. The basis of coral reefs relies on cnidarian-dinoflagellate endosymbiosis [3]. It is critical to understand the mechanism of symbiotic relationships to help preserve or improve the health of coral reefs.

Corals have many different types of cells, including ciliated column, secretory, cnidocyte, flagellated columnar, cuboidal, absorptive, squamous, anchoring, flagellated cuboidal, spindle-shaped, sensory, motor neuron, interneuron, and neurosecretory cells [4,5]. The most critical cells for cnidarian-dinoflagellate endosymbiosis are the host cells that house algal cells, also called symbiotic gastrodermal cells (SGCs). SGCs exist in the endoderm of cnidarians and in the main cells where symbiotic relationships occur. Many studies have investigated the mechanism of cnidarian-dinoflagellate endosymbiosis using SGCs, and it would be necessary to first isolate SGCs from stony corals (*Euphyllia glabrescens*) at the beginning of such studies. To do so, researchers have explored two different methods [6,7]. The function of the SGC membrane has also been discussed. Some studies have found that SGC membrane trafficking increases the susceptibility of algal symbionts to photo-inhibition [8], and SGC membranes are involved in the recognition and phagocytosis of *Symbiodiniaceae* [9,10]. The proteomic and ultrastructural of the lipid body in SGCs have also been studied by Peng et al. [11]. Our group studies the proteomics of SGCs. We isolated SGCs from stony corals (*Euphyllia glabrescens*) and labeled them using biotin-XX sulfosuccinimidyl ester to study the cell surface proteins. We identified 19 proteins that involved cnidarian-dinoflagellate endosymbiosis [12]. Furthermore, we used SGCs as an antigen to generate antibodies and produced a new monoclonal antibody that specifically recognizes clade C (now renamed *Cladocopium*) [13] symbionts, but not their free-living counterparts or other *Symbiodiniacea*. The monoclonal proteins recognizable by the antibody are symbiotic markers for *Cladocopium* symbionts, whose expression levels vary depending on the health status of corals [14]. Accordingly, SGC membranes may regulate the stability of the host coral and dinoflagellates. SGCs also play a vital role in cnidarian-dinoflagellate endosymbiosis.

The discussion on the endosymbiotic mechanism of cnidarians and dinoflagellates has mostly involved stony corals (e.g., *Euphyllia glabrescens*) [11,15,16,17,18] or sea anemones (e.g., *Exaiptasia pallida*) [19,20,21,22] as the animal models. Sea anemones have often been used as the model species in symbiosis research because they can be completely bleached and survive without algal cells through heterotrophy [20,22,23] (feeding on brine shrimp, *Artemia*). No studies have mentioned the endosymbiotic mechanism of octocoral and dinoflagellates. For years, people have thought that stony corals have immense skeleton- and reef-building capabilities and that their skeletons form an important habitat for coral reef organisms, while octocorals do not. Therefore, the importance of octocorals has been neglected. However, in a 2011 study, it was confirmed that octocorals (*Sinularia*) have the ability to cement sclerites and consolidate them at their base into spiculite, thus making them reef builders [24]. This study deepens the understanding of the importance of octocorals in coral reef ecosystems. It is evident that octocorals also play an important role in the entire coral reef ecosystem. In addition, octocorals are the second most common benthic component on many shallow reefs and a major component on deep reefs (mesophotic coral-reef ecosystems) in the Red Sea [25]. However, research on octocorals-dinoflagellate endosymbiosis is very rare, and no one has used octocorals as an animal model to study endosymbiosis. Scleractinian coral and octocoral are two different types of corals. We aim to compare these two and study the octocorals-dinoflagellate endosymbiosis by incorporating our experiences with the scleractinian corals. As mentioned above, SGCs play a vital role in cnidarian-dinoflagellate endosymbiosis, and collecting SGCs to study the cnidarian-dinoflagellate endosymbiosis is required. However, as we tried to collect SGCs from the octocoral *Sinularia flexibilis* following Khalesi’s paper, we found the amount of intact SGCs was very small. Therefore, we developed a quick and easy method for collecting large amounts of SGCs from octocorals. We hope that this method can help researchers pay closer attention to octocorals and study SGCs with more ease.

## 2. Results and Discussion

### 2.1. Incubation of Octocorals in a High-Salinity Solution Increased the Amount of SGCs Released from the Tentacles

In our previous works, we successfully isolated SGCs from *E. glabrescens* and studied the endosymbiosis relationship between stony corals and dinoflagellates [6,12,14]. We now wanted to establish a new model system using octocorals as an animal model to study the endosymbiotic relationship between octocorals and dinoflagellates. Our first step was to collect the SGCs from octocorals. In our husbandry center, many cultures of the octocoral *S. flexibilis* are available, and many compounds isolated from *S. flexibilis* are potential drugs. For this reason, we chose *S. flexibilis* as our experimental organism. Initially, we used the methods mentioned by Khalesi [26] to dissociate cells from *S. flexibilis*, but only a small amount of intact SGCs were observed (less than 5%), while host cells and algal cells accounted for the majority of the dissociated cells (more than 90%). It was difficult for us to collect a sufficient amount of SGCs for the experiments. To solve this problem, we attempted many ways to obtain more intact SGCs from *S. flexibilis*. After testing the various methods over one year, we found that incubating the tentacles of *S. flexibilis* overnight in high-salinity (60‰) filtered seawater (FSW) resulted in many SGCs auto-releasing from the tentacles (Figure 1). This finding provided us with a new approach for collecting SGCs from octocorals.

### 2.2. Salinity and Osmotic Pressure Affect the Ratio of SGCs Released in S. flexibilis

To explore the effect of salinity on the SGC ratio, the tentacles of *S. flexibilis* were incubated overnight in FSW of different salinities (from 36‰ to 75‰, salinity adjusted by adding sodium chloride), and the ratio of SGCs was quantified using Photoshop. The ratio of SGCs under the different salinities had a standard normal distribution that peaks at 60‰ (Figure 2A). When the salinity was below 55‰, the ratio of SGCs increased with higher salinity. When the salinity was higher than 60‰, the ratio of SGCs obviously decreased. When the salinity reached 75‰, less than 1% SGCs was observed. A salinity higher than 60‰ may cause cell damage and broken SGC membranes in *S. flexibilis*, leading to the release of symbiotic algae. We also performed the same experiment using potassium chloride to adjust the salinity and obtained a pattern similar to that of sodium chloride (data not shown). This means that sodium ions are not the key point; rather, it is salinity/osmotic pressure that matters. Therefore, we measured the osmotic pressures of the solutions of varying salinity adjusted by either sodium chloride or potassium chloride. As we expected, the curves of sodium chloride and potassium chloride overlap almost completely (Figure 2B). This result implies that the intracellular osmotic pressures in SGCs of *S. flexibilis* may be closer to 1800 mOsm and not the normal seawater osmotic pressure (approximately 1000 mOsm).

### 2.3. Time Affects the Ratio of SGCs Released from S. flexibilis

Although we identified a new method for collecting large amounts of SGCs, this method was time consuming (more than 16 h). To increase the experimental efficiency, we attempted to reduce the incubation time. The tentacles of *S. flexibilis* were incubated in 36‰ or 60‰ FSW over different time courses (2 to 16 h), and the ratio of SGCs was counted. When the tentacles of *S. flexibilis* were incubated in 60‰ FSW for only 2 h, the ratio of SGCs was fivefold higher than what the 36‰ FSW produced. The ratio peaked at 6 h (Figure 3). The incubation of *S. flexibilis* tentacles in 60‰ FSW for 6 h resulted in a greater amount of SGCs (approximately 18-fold higher) than what the 36‰ FSW produced. Using this improved method, researchers can easily collect large amounts of SGCs from octocorals in a short period of time (2 to 6 h).

### 2.4. Salinity Affects Tissue Morphology in S. flexibilis

To addressed why higher salinity increased the amount of SGCs released from *S. flexibilis*, we performed hematoxylin & eosin (H&E) staining to observe the tissue morphology of the tentacles incubated 36‰ or 60‰ FSW for 6 h. The results showed that many cells detached in the 60‰ FSW (indicated by the black circles in Figure 4), but they attached tightly in the 36‰ FSW (indicated by the black arrows). These detached SGCs were easily released when the tentacles were cut into small pieces. This may have been caused by the high-salinity solution breaking the cell–cell tight junction and leading to cell detachment from tissues, but the detailed mechanism needs further exploration. Other than the amount of detached SGCs, we have not observed any other differences between these two tissues. Therefore, we conclude that high salinity allows SGCs to detach from the tissue more easily, resulting in a more efficient method for collecting SGCs.

### 2.5. The Viability of Cells Treated with 60‰ FSW in S. flexibilis

To determine whether the detached cells in the 60‰ FSW were damaged by salt stress, we performed three assays to test their cell viability. First, we incubated the detached cells in fluorescent dextran to detect the membrane intactness. If the cell membrane is damaged, fluorescent dextran will enter the cell, causing the cell cytosol to fluoresce green, as seen in the positive image at the lower right corner of Figure 5A. In contrast, if the cell membrane is intact, then the cytosol will be dark and without any fluorescence. In Figure 5A, the images at the upper right and middle right positions show that all of the SGC cytosols were dark, demonstrating that the SGC membrane was intact after incubating in high-salinity FSW. The green spots in the symbiotic algae in Figure 5A were caused by the autofluorescence of the accumulation body and not by the dextran. Second, we used the 3-(4,5-Dimethylthiazol-2-yl)-2,5-diphenyl tetrazolium bromide (MTT) assay to test the mitochondrial activity of the detached cells in 60‰ FSW. Unexpectedly, the viability of cells treated with 60‰ FSW was not reduced; rather, it was higher than that under the 36‰ FSW (Figure 5B). High salinity not only does not reduce cell viability, but increases cell viability approximately twofold. We repeated the experiment three times, and all replicates provided the same results. Therefore, we performed a third assay to confirm cell viability. We detected the adenosine triphosphate (ATP) activity of the detached cells in the 60‰ FSW and the 36‰ FSW. Cell death (especially apoptosis) is a process that consumes energy, so ATP activity was used as an indicator of cell viability. When cell death occurs, ATP activity decreases because dead cells no longer perform basic metabolic functions. As predicted, we obtained results similar to those in the MTT assay. The ATP activity of cells treated with 60‰ FSW was higher than that of cells treated with 36‰ FSW (Figure 5C). In summary, the results from the above three analyses (Figure 5A–C) confirm that the detached cells are intact and that the cell viability is good after incubating in a high-salinity solution.

In a previous study [26], Khalesi tried three ways to dissociate cells from *S. flexibilis* and used different media to culture these cells, including coral cells and symbionts. When he used a mechanical dissociation process, he observed some coral cells with their endosymbionts (what we call SGCs), but these were very low in number in his Figure 3B. In his observations, he did not mention the ratios of different types of cells. According to his Figure 3C, most cells were zooxanthellae cells, few were coral cells, and SGCs were even harder to find. In his experiment, some conditions could produce large amounts of cells, but the cells did not grow. In our experiment, the viability of cells treated with the 60‰ FSW was twofold higher than that of those treated with the 36‰ FSW (Figure 5B). This new method can increase cell viability and provides a new approach for culturing coral cells, zooxanthellae cells, or SGCs from octocorals.

The secondary metabolites isolated from *S. flexibilis* had many different functions, including anti-microbial [27], anti-cancer [28,29,30,31,32,33,34], and anti-inflammatory [35,36,37,38] activities. *S. flexibilis* has the potential for drug development [39]. Large-scale cultivation of *S. flexibilis* or cultured cells is the future trend. As *S. flexibilis* is a producer of potential pharmaceuticals, and sustainable mass production of these corals represents a possible source, the culture of coral cells from *S. flexibilis* is a pressing need.

### 2.6. Salinity and Time Affect the Ratio of SGCs Released from P. thyrsoides and S. compressa

All of the above results apply to *S. flexibilis*, but it was unclear whether this method for collecting large amounts of SGCs applies only to *S. flexibilis* or if it can be used for other octocorals. We performed the same experiment on two species of octocorals, *P. thyrsoides* and *S. compressa*. As we expected, high salinity increased both the number and the ratio of SGCs released from these two octocorals (Figure 6 and Figure 7). For *P. thyrsoides*, the highest ratio of SGCs was released from the tentacles treated with 50‰ FSW for 6 h (Figure 7A,B). For *S. compressa*, the highest ratio of SGCs was released from the tentacles treated with 55‰ FSW for 16 h (Figure 7C,D). Although different types of octocorals have different optimal conditions, high salinity can indeed increase the ratio of SGCs released from these three octocorals. We also performed the same experiment with two scleractinian corals, *E. glabrescens* and *Pocillopora damicornis*, but no SGCs were observed, and the tentacles rotted after incubating in the high-salinity solution (data not shown). Therefore, a high-salinity-induced increase in the SGC ratio is a common phenomenon among the three octocorals, but not the scleractinian corals. This result also implies that the octocorals and scleractinian corals may have different regulation mechanisms for osmotic pressure or different tolerance levels for osmotic pressure changes. The maximum number of SGCs from the three octocorals came from the tentacles treated with 50‰–60‰ FSW; these results implied that the intracellular osmotic pressures in the SGCs of *S. flexibilis*, *P. thyrsoides*, and *S. compressa* may be closer to 1500–1800 mOsm rather than the normal seawater osmotic pressure (approximately 1000 mOsm). Current technology cannot measure intracellular osmotic pressures in SGCs, but this special phenomenon is worthy of further research and future discussions.

All cells must constantly adjust their cell volume to maintain the equilibrium between intracellular and extracellular osmolarity, a process known as osmoregulation [40]. In SGCs, osmoregulation is complicated. The host must balance its extracellular osmolarity with an intracellular environment that is influenced by both its own metabolism and that of its symbionts. Our results imply that the osmoregulation in octocorals obviously differs from that in scleractinian corals. The endosymbiotic relationship in octocorals and scleractinian corals may also be different. It has been demonstrated that symbiotic zooxanthellae live within an osmotically different environment from that of free-living dinoflagellates [41]. Previous studies have demonstrated that high salinity induces glycerol production in cultured *Symbiodiniaceae* [42] and that high salinity also induces *Symbiodiniaceae* to produce high levels of floridoside both in vitro and *in hospite* [43]. Therefore, osmoregulation may be affected by cnidarian-dinoflagellate endosymbiosis. The detailed mechanism is worthy of further study.

## 3. Materials and Methods

### 3.1. Reagents and Culture Media

All chemical reagents were purchase from Merck KGaA, Darmstadt, Germany. The FSW was prepared by filtering natural seawater through a filter unit (0.2 µm pore size, BF500-02-24, Biofree, Taipei, Taiwan). Various FSW salinities between 36‰ and 75‰ were obtained by adding different amounts of sodium chloride or potassium chloride, and the salinity was measured with a refractometer (Land Agri Co., Kaohsiung, Taiwan).

### 3.2. Coral Collection and Maintenance

*S. flexibilis*, *P. thyrsoides*, and *S. compressa* were cultured in the husbandry center of the National Museum of Marine Biology and Aquarium for many years. The corals used in this study were maintained under a natural photoperiod with additional air circulation in a 4-ton outdoor aquarium with flow-through seawater. The temperature in the tank was maintained at 27 ± 1 °C by a microprocessor-controlled cooler (IPO-500, 1/3HP, TungFa Co., Keelung, Taiwan).

### 3.3. Quantifying the Ratio of SGCs

Approximately 1 cm of the tentacles was amputated from the octocorals using curved surgical scissors. The tentacles of the octocorals were immersed in 24-well plates containing various salinities of FSW for 16 h. The tentacles were rinsed with FSW and then cut into small pieces with surgical scissors to release whole cells, including host cells, SGCs, and *Symbiodiniaceae* cells. These cells were filtered with a 70 µm cell strainer to remove any large tissue fragments, and the filtrate was collected in a centrifuge tube. The filtrate was centrifuged at 700 rpm for 5 min, the supernatant was discarded, and the pellet was resuspended in FSW. The cells were ready to be observed and photographed under a light microscope. More than 20 photographs were randomly captured under various salinities of FSW. Using a light microscope, we could see three types of cells. The white transparent cells were the host cells, the brown cells were the *Symbiodiniaceae* cells, and the brown cells living inside the white cells were the SGCs. The numbers of SGCs and *Symbiodiniaceae* cells were counted using Adobe Photoshop CS6, and host cells were not included in the calculations (the number of the host cells was much higher than the numbers of SGCs and *Symbiodiniaceae* cells). Under each condition, more than 1000 cells were counted, and the percentage of SGCs was calculated. Photoshop can count the number of cells and label serial numbers automatically. Any errors were fixed manually to get the correct number of SGCs and *Symbiodiniaceae* cells. The ratio of SGCs was calculated by dividing the number of SGC by the sum of SGCs and *Symbiodiniaceae* cells.

### 3.4. Hematoxylin & Eosin Staining (H&E Staining)

The tentacles of *S. flexibilis* were incubated in 36‰ or 60‰ FSW for 6 h and then were fixed in 4% paraformaldehyde/FSW overnight at 4 °C. The tissues were embedded in a paraffin block and cut into 5 μm thick sections. The paraffin was removed from the sections following the processes described in a previous study [16]. The sections were stained with Gill 2 hematoxylin (Thermo Fisher Scientific, Waltham, MA, USA) and washed with water. The sections were rinsed multiple times in water-ethanol mixtures. With each rinse, the prepared mixture contained a higher ethanol-to-water ratio than the previous one, and the last rinse contained only ethanol. After the sections were stained with eosin Y (Thermo Fisher Scientific), they were washed with ethanol, and the slide was dipped into xylene. One or two drops of Histomount (Electron Microscopy Sciences, Hatfield, PA, USA) were added, and the slides were covered with a coverslip.

### 3.5. Viability Assay

The tentacles of *S. flexibilis* were incubated in 36‰ or 60‰ FSW in 24-well plates for 6 h, cut into small pieces, filtered with cell strainers, and centrifuged at 100× *g* for 5 min in centrifuged tubes to collect the cells. These two groups of samples were prepared freshly. Then, three viability assays were performed.

#### 3.5.1. Cell Intactness Assay

Dextrans are hydrophilic polysaccharides with good water solubility and low toxicity. Fluorescent dextrans can serve as markers of cell intactness. In this study, fluorescent dextrans with a 10,000-dalton molecular weight (molecular probe Cat. No. D22910) were used to check the cell intactness. We added water to prepare a dextran stock solution (10 mg mL^−1^) that was filtrated by a 0.2 μm pore-diameter sterile filter and stored at −20 °C protected from light. To use it, we added 2 μL of the dextran stock solution into 100 μL of cells and incubated for 10 min protected from light. Then, we observed some of the cells under fluorescence microscopy. When the cell membrane has a small break, the fluorescent dextrans will enter the cells, and the cytosol will fluoresce green. If the cell is intact, the fluorescent dextrans cannot enter the cell, and the cytosol is dark (black). The cells released from the octocorals were incubated in fluorescent dextrans following the manufacturer’s instructions.

#### 3.5.2. Mitochondrial Activity Assay

Mitochondrial activity was determined with a 3-(4,5-dimethylthiazol-2-yl)-2,5-diphenyl tetrazolium bromide (MTT) assay. MTT is a yellow substrate in the mitochondria, where it can be cleaved into formazan, a purple insoluble compound. This reaction occurs only in active mitochondria, while dead cells with dysfunctional mitochondria do not produce formazan. An MTT stock solution (5 mg mL^−1^) was prepared by adding MTT powder to FSW and stored at 4 °C in the dark. Two groups of cells (36‰ and 60‰) were prepared freshly as described in Section 3.5. Each sample was triplicated; 2 × 10^6^ cells per well were seeded into a 96-well culture plate; 100 µL of MTT stock solution was added; the plate was kept in the dark for 4 h. Centrifuged at 3000× *g* for 5 min by a plate centrifuge to collect the cells, the MTT solution was then discarded, and the cells were washed with 200 µL of FSW to remove the residual MTT. Next, 200 µL of a solvent (2N potassium hydroxide/dimethyl sulfoxide = 1:1 (*v*/*v*)) was added, and the solution was shaken until the formazan crystals completely dissolved. Then, 100 µL of the purple solution was removed, and the absorbance was detected at 570 nm with a reference background of 630 nm by a Microplate Reader (Synergy H4 with Hybrid Technology™, BioTek, Winooski, VT, USA), followed by normalization with the protein concentration. Each assay was performed in triplicate.

#### 3.5.3. Adenosine Triphosphate (ATP) Activity Assay

The ATP activity was measured using a kit (ApoSENSOR^TM^ ATP Cell Viability Assay Kit, BioVision, Milpitas, CA, USA) according to the manufacturer’s instructions and normalization with the protein concentration. In brief, two groups of cells (36‰ and 60‰) were prepared freshly as described in Section 3.5. Each sample was triplicated. The 1 × 10^4^ cells per sample were transferred into a luminometer tube (Polystyrene, 12 × 75 mm, Ming-Jie Co., Taichung, Taiwan) and 100 μL of nuclear releasing reagent was added at room temperature with 5 min of gentle shaking. Next, 1 μL of ATP monitoring enzyme was added into the cell lysate. The sample was read after 1 min in the luminometer (Lumat LB 9507 Ultra-Sensitive Tube Luminometer, BERTHOLD TECHNOLOGIES GmbH & Co KG, Bad Wildbad, Germany). The same amount of cells (1 × 10^4^ cells) were also broken by a homogenizer to obtain the total protein count. Detail information can be found at https://www.biovision.com/documentation/datasheets/K254.pdf. Each assay was performed in triplicate.

### 3.6. Measurement of Osmolality

The osmotic pressure of the solutions of various salinity was measured using a micro-osmometer (Model 3300, Advanced Instruments, Norwood, MA, USA) according to the manufacturer’s instructions.

### 3.7. Statistical Analysis

The experimental data were examined using GraphPad Prism (GraphPad Software Inc., San Diego, CA, USA). *p* < 0.05 was considered statistically significant.

## 4. Conclusions

We developed a new method to collect large amounts of SGCs from octocorals. This method can be successfully used for three species of octocorals: *S. flexibilis*, *P. thyrsoides*, and *S. compressa*. A flow chart of this method is shown in Figure 8. We believe this method is also applicable to other octocorals. The SGCs of octocorals can be used to study the endosymbiotic mechanism between octocorals and dinoflagellates. We can study the functions, characteristics, and structures of octocoral SCGs using proteomics, transcriptomics, lipidomics, metabolomics, and multiomics. We can also try to culture octocoral SGCs. This method serves as a good tool for research related to coral reefs.

## Figures and Tables

**Figure 1 ijms-21-03911-f001:**
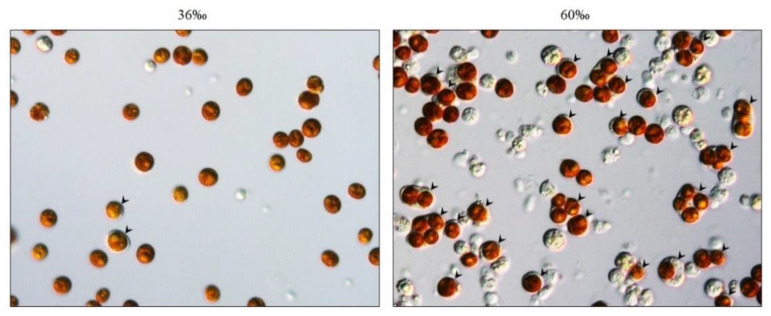
A large amount of symbiotic gastrodermal cells (SGCs) were released from *S. flexibilis* tentacles incubated in 60‰ filtered seawater (FSW). The tentacles of *S. flexibilis* were incubated overnight in the control FSW (36‰) and the high-salinity FSW (60‰), and were then cut into small pieces. The released cells were collected and observed via microscopy (400×). The arrowheads indicate the SGCs.

**Figure 2 ijms-21-03911-f002:**
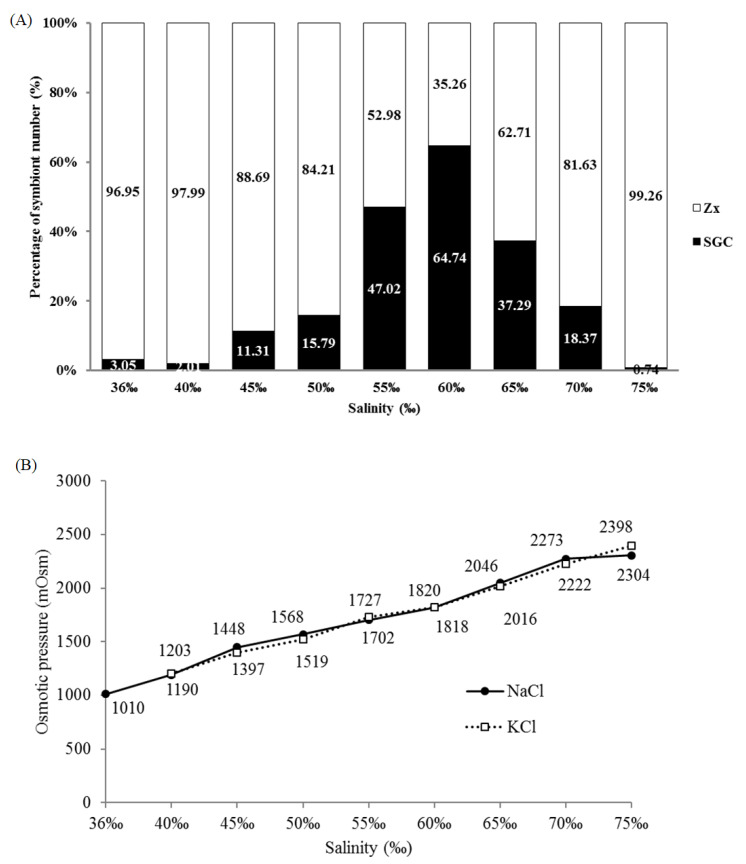
Salinity and osmotic pressure affect the ratio of SGCs released by *S. flexibilis*. (**A**) Tentacles of *S. flexibilis* were incubated overnight in FSW of different salinities. The ratios of SGCs associated with the different salinities show a standard normal distribution, with the peak at 60‰ for *S. flexibilis*. Zx: zooxanthellae. (**B**) The relationship between salinity and osmotic pressure. The various salinities of FSW were obtained by adding different amounts of sodium chloride (NaCl) or potassium chloride (KCl), and these two curves overlap almost completely.

**Figure 3 ijms-21-03911-f003:**
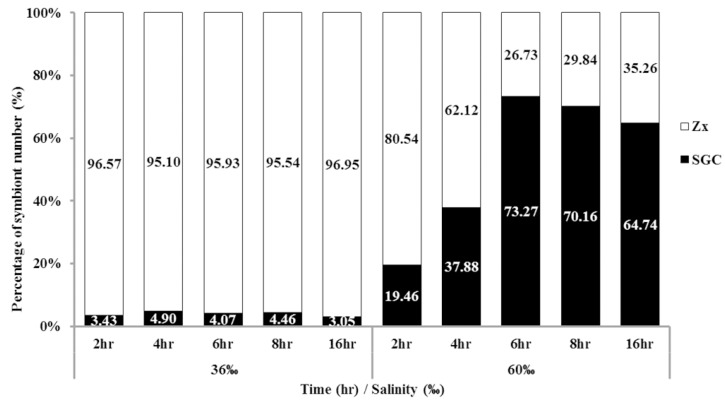
Time affects the ratio of SGCs released from *S. flexibilis*. The tentacles of *S. flexibilis* were incubated in 36‰ or 60‰ FSW over different time courses. The peak ratio of SGCs was at 6 h. Zx: zooxanthellae.

**Figure 4 ijms-21-03911-f004:**
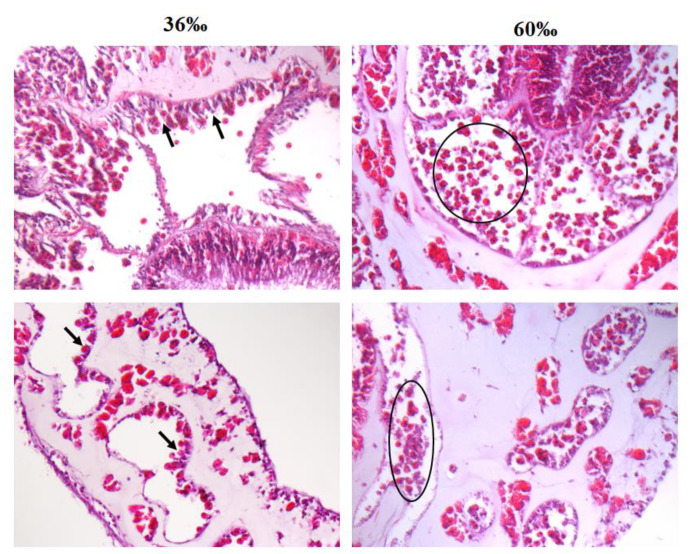
Salinity affects the tissue morphology in *S. flexibilis*. The tentacles of *S. flexibilis* were incubated in 36‰ or 60‰ FSW for 6 h, and the tentacles were processed with hematoxylin & eosin (H&E) staining. The black arrows in the left panel indicate the attached SGCs in the 36‰ FSW, and the black circles in the right panel indicate the detached SGCs in the 60‰ FSW. These slides were observed via microscopy (400×).

**Figure 5 ijms-21-03911-f005:**
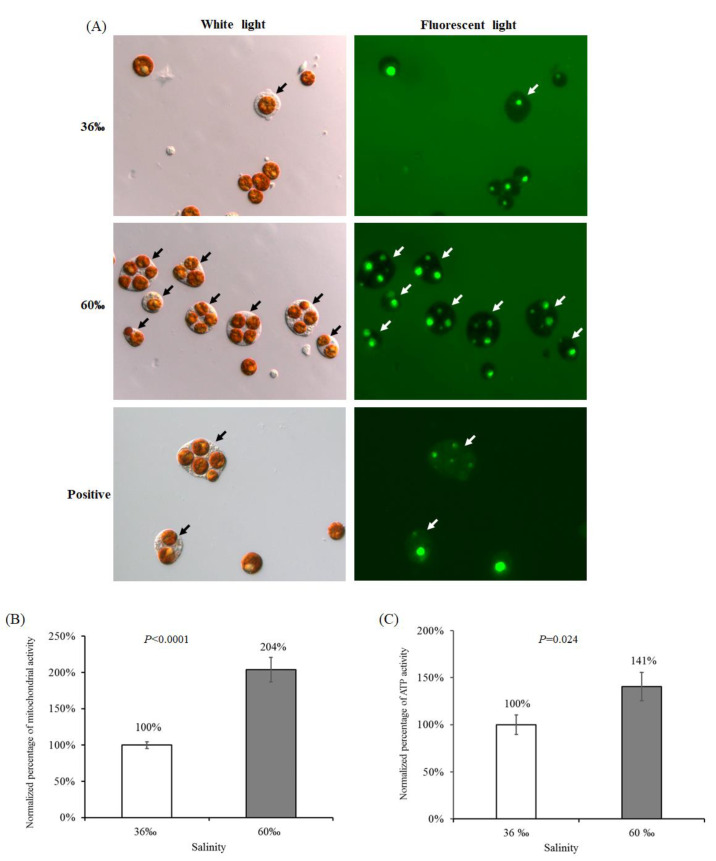
The viability of *S. flexibilis* cells treated with the 60‰ FSW. (**A**) The released cells were stained with fluorescent dextran and observed via microscopy (400×). The black arrow indicates the SGCs under white light. The white arrows indicate the SGCs under fluorescent light, and the dark cytosol indicates that the SGCs are intact. Positive means positive control, which were the released cells treated with 0.00001% Triton X-100 for 10 min to damage their cell membranes. (**B**) 3-(4,5-Dimethylthiazol-2-yl)-2,5-diphenyl tetrazolium bromide (MTT) assay and (**C**) adenosine triphosphate (ATP) activity show that the cells released from the tentacles incubated in the 60‰ FSW had higher cell viability than those in the control treatment.

**Figure 6 ijms-21-03911-f006:**
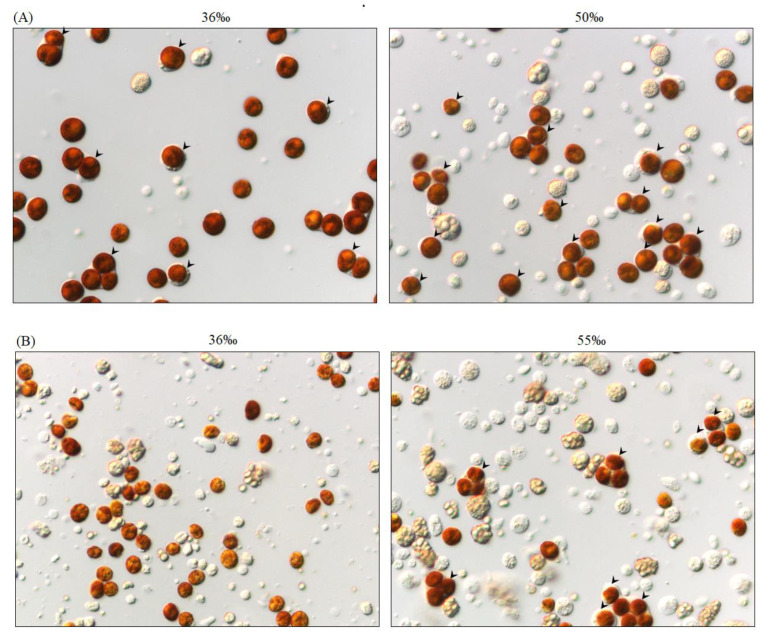
High salinity increased the number of SGCs released from *P. thyrsoides* and *S. compressa*. The tentacles of (**A**) *P. thyrsoides* and (**B**) *S. compressa* were incubated overnight in the control FSW (36‰) and high-salinity FSW (50‰ or 55‰) and then cut into small pieces. The released cells were observed via microscopy (400×). The arrowheads indicate the SGCs.

**Figure 7 ijms-21-03911-f007:**
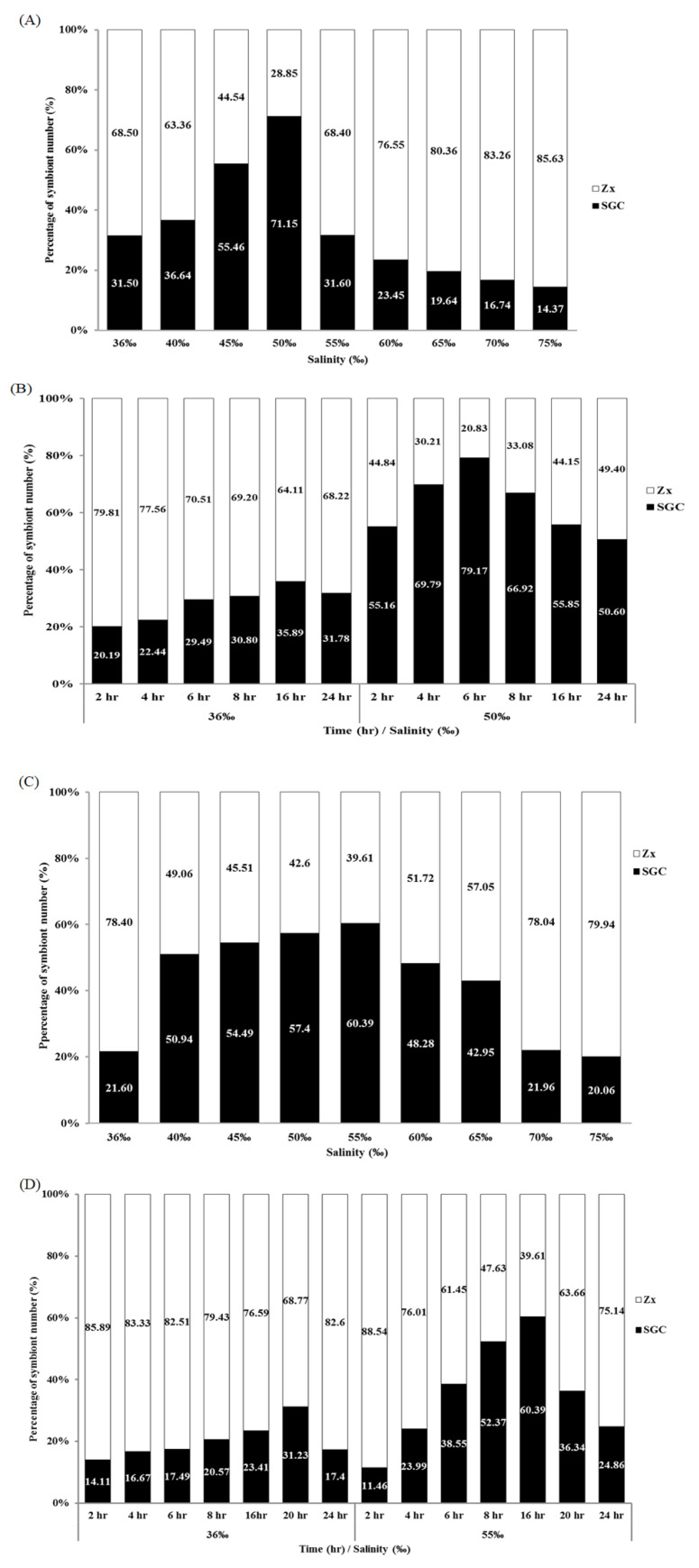
Salinity and time affect the ratio of SGCs released from *P. thyrsoides* and *S. compressa*. The tentacles of (**A**) *P. thyrsoides* and (**C**) *S. compressa* were incubated overnight in FSW of different salinities. The tentacles of (**B**) *P. thyrsoides* and (**D**) *S. compressa* were incubated in 50‰ or 55‰ FSW over different time courses. Zx: zooxanthellae.

**Figure 8 ijms-21-03911-f008:**
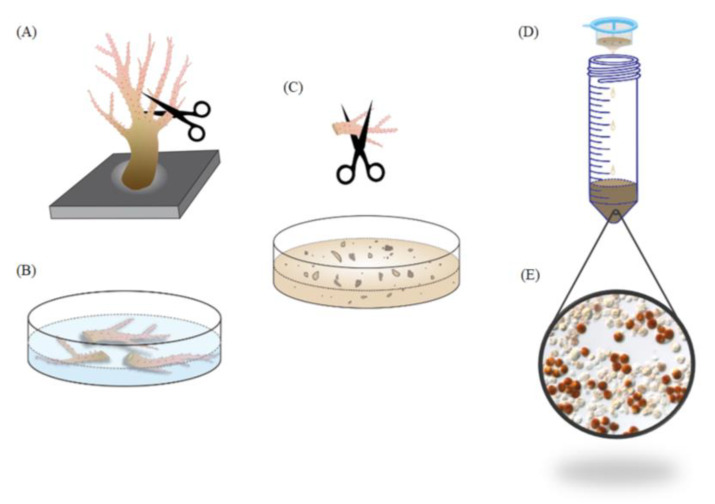
Flow chart of the new method for collecting large amounts of SGCs from octocorals. (**A**) Cut some tentacles from soft corals and (**B**) immerse them in high-salinity FSW for a few hours. (**C**) Select one tentacle and cut it into small pieces of approximately 1–2 mm. (**D**) Filter this solution with a cell strainer to remove any large tissue fragments. Centrifuge the filtered solution and wash the cells with normal FSW, and (**E**) large amounts of SGCs become available.

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
