# Peer review of "A New Method for Collecting Large Amounts of Symbiotic Gastrodermal Cells from Octocorals"

_ijms, 2020, doi:10.3390/ijms21113911_

Round 1

Reviewer 1 Report

The manuscript presented by Chiu et al. describes a new improved method to isolate gastrodermal cells from the rarely investigated octocorals. The authors show some evidence that the symbiotic gastrodermal cells could be isolated with higher efficiency and the cells exhibited significant viability, therefore the method and the findings of the study could be an important methodological development in coral research. However, in its current form, the manuscript cannot be recommended for publication, because there are severe shortcomings and lack of essential details on the applied methodology and the experimental design.

The manuscript also requires a significant and thorough revision in English grammar and style.

Specific comments:

line 56-58: the mentioned cell types could not be identified in the cited source. Add full details to ref. 11

line 67-69: Rephrase, authors should address the aims more clearly, not only making attention about the importance of SGC in octocorals, but for what aims SGCs might be important and what questions might be answered.

line 73 and all other relevant sections: species names should be in italics.

line 77: unclear term ’museum’ – does it refer to the coral collection?

line 82-83: many ways of obtaining intact SGCs are mentioned, however it remains unclear what were these methods exactly, authors should add more information. It is also unclear what does ’long-term test’ refer to.

line 95-96: it is unclear how was the ratio of SGCs quantified by the software, add the relevant details.

line 102 – have the authors attempted to investigate the effect of the valence of the various ions? If osmotic pressure might be a key point, have the authors attempted to apply sorbitol, mannitol or other, frequently applied sugaralcohols to experimentally adjust osmotic pressure? It would be important to consider various experimental conditions to optimize SGC release.

line 130 and other sections: some rephrasing in style might be necessary - 'We wanted to know...' etc could be rephrased, e.g. we addressed the question, or similar.

Figure 5: why is the background in the 'positive' samples dark, and in the other treatments bright green? How were the samples treated after labeling, e.g. how was washing done? Authors should also show and describe the statistical analysis of the fluorescence analysis, and quantify the percentage of the intact vs. broken SGCs in the different treatments. Methodological details on the fluorescent dextran staining is missing (as indicated below). The results on MTT and ATP assays are peculiar, could the elevated mitochondrial activity and ATPase activity indicate elevated stress instead of elevated viability? As these results are questionable, it is essential that the authors cross-validate their viability assays with other methods, such as intracellular esterase assay, photosynthetic activity etc.

Section 3.5.1 – this methodological description is incomplete as it is just the repetition of the principle of the method. Authors should describe how did they optimize the method to SGCs instead of just generally saying that they followed manufacturers instructions. the details of the microscopic analysis have to be given. After labeling, were the cells washed and how? More details about the fluorescent dextran labeling method has to be added.

Section 3.5.2. – a thorough clarification of the applied method has to be completed. The arrangement of the experiment should be described in details. If plate assays were applied, the experimental design should indicate how were the samples arranged for plate assay and how were the samples treated in this matrix arrangement.

line 282: how was this step done? in a plate centrifuge?

line 284: were these steps done repeatedly in the 96 well plate?

line 285: How was absorbance measured? By a spectrophotometer? using a plate reader?

Section 3.5.3 – this section is incomplete, therefore the presented results on the ATPase activity cannot be judged.

Reviewer 2 Report

A new method for collecting large amounts of symbiotic gastrodermal cells from octocorals authored by Chiu et al. describe a new method to isolate symbiotic gastrodermal cell from octocoral cells. One unique finding of this research is that they found incubating the tissue in high salinity resulted in an increased ratio of SGCs for three octocoral species but not for hexacoral species. 

Overall the findings of this research are intriguing and worthwhile publishing. However, the manuscript still needs some modifications before publication.

1) It was unclear how the isolated SGCs can be used to study endosymbiotic mechanisms of octocorals. Please describe in more details in the introduction so that authors can put emphasis on the importance of their findings of the research.

2) Please cite some previous research that described the importance of studying endosymbiotic mechanisms  (if any, octocoral or other species)in the second paragraph of the introduction.

3) L 54 it was not clear and seemed out of the blue to say "To study the endosymbiotic mechanisms of octocoral and dinoflagellates is important." Please add more information as why it is important

4) all the scientific names should be in Italic. Please change

5) L84 Please describe how authors identify SGCs from other cells under the light scope in at least material and method section.

6)Please write the full original words when you use abbreviations for the first time (e.g. H & E, FSW, MTT etc).

7)L138 grammatical error

8)L175 Since S. flexibilis is a producer.... move these information to introduction. It also needs a few citations.

9)L198 Because authors only tested three species of octocorals, it is still risky to say "a high-salinity-induces increased ratio of SGCs is a common phenomenon". would be a common phenomenon though.

10) Fig. 7 (B) S. compressa were...→(C) S. compressa werer

(C) P. thyroides and → (B)P. thyroides and

11)Material and Method section please add more detailed information about the experimental instrument and chemical, filter was used (manufacture's name and product names. e.g. fileter, microprocessor-controlled cooler,

12)L289 performed→measured?

13) P should be in italic.

14) Again please discuss why examining endosymbitic mechanisms between octocorals  and dinoflagellate using the current is so important and how this new method can be used for that purpose.

Round 2

Reviewer 1 Report

The manuscript has improved significantly after revision, therefore it can be accepted for publication.

However, the revised version of the manuscript still contains a couple of grammatical errors. For example in line 146, 'To addressed' is incorrect, 'To address' or 'In order to address' would be more appropriate.

Therefore, another check of English grammar might be necessary.